# New Design of a Supervised Energy Disaggregation Model Based on the Deep Neural Network for a Smart Grid

**İsmail Hakkı ÇAVDAR and Vahid FARYAD \***

Department of Electric-Electronic, Karadeniz Technical University, 61080 Trabzon, Turkey; cavdar@ktu.edu.tr
**\*** Correspondence: vahit.feryat@gmail.com; Tel.: +90-53-5733-1609

**Abstract:** Energy management technology of demand-side is a key process of the smart grid that helps achieve a more efficient use of generation assets by reducing the energy demand of users during peak loads. In the context of a smart grid and smart metering, this paper proposes a hybrid model of energy disaggregation through deep feature learning for non-intrusive load monitoring to classify home appliances based on the information of main meters. In addition, a deep neural model of supervised energy disaggregation with a high accuracy for giving awareness to end users and generating detailed feedback from demand-side with no need for expensive smart outlet sensors was introduced. A new functional API model of deep learning (DL) based on energy disaggregation was designed by combining a one-dimensional convolutional neural network and recurrent neural network (1D CNN-RNN). The proposed model was trained on Google Colab's Tesla graphics processing unit (GPU) using Keras. The residential energy disaggregation dataset was used for real households and was implemented in Tensorflow backend. Three different disaggregation methods were compared, namely the convolutional neural network, 1D CNN-RNN, and long short-term memory. The results showed that energy can be disaggregated from the metrics very accurately using the proposed 1D CNN-RNN model. Finally, as a work in progress, we introduced the DL on the Edge for Fog Computing non-intrusive load monitoring (NILM) on a low-cost embedded board using a state-of-the-art inference library called uTensor that can support any Mbed enabled board with no need for the DL API of web services and internet connectivity.

**Keywords:** smart grid; deep neural networks; non-intrusive load monitoring; supervised energy disaggregation; deep feature learning; tensor flow; GPU; uTensor

---

## 1. Introduction

Commercial and residential buildings consume about 60% of electricity in the world. For instance, buildings use 74.9% of the generated electricity in the United States of America, and this figure in Africa is 56%. It is estimated that by 2050, the energy demand by construction sector and use of energy in roofed spaces will increase by around 80%. These figures highlight the potential of reducing the energy demand through frugal energy use in buildings [1]. Effective frugal energy use in residential buildings can be accomplished through real time (RT) monitoring of the energy consumption of electrical appliances, providing RT feedbacks to end users to improve their awareness of what electrical appliances should be used at specific times, how much, and the reason for energy consumption by electrical appliances [2]. In this way, end users have the opportunity to play an active role in controlling and preventing energy waste. In addition, the way in which they contribute to the frugal use of energy will be a measure of their motivation to save energy. Studies have shown that awareness about energy consumption, along with RT feedback to a household, create positive changes

and lead end users toward sustainable energy use [3]. In general, instant energy usage of electrical appliances can be determined using smart outlet sensors that determine the energy consumption of each appliance. Such devices are expensive and demand unique communication protocols [4].

Currently, the use of smart sensors at a large scale has drawn attention to the development of NILM methods [5]. These methods refer to computational methods where aggregated data of energy usage from a unique source of data like a smart sensor is used to determine the energy use of every single electrical appliance in the house. These methods guarantee saving household costs through implementing RT monitoring of frugal energy consumption of electrical appliances. They also provide the chance to preserve and save energy. Moreover, the NILM system helps policy makers to measure the success of their energy performance strategies and predict energy demands. In this way, the suppliers can plan to supply the demands in an optimum way [6]. The NILM concept is decades old; however, recently, we have seen a trending interest in this field of research inspired by parallel advancements in data communications, networks, sensing technology, machine learning, GPUs, and deep learning methods. The NILM is a principal prerequisite for identifying appliances and providing energy feedback to the residential consumers; however, it is equally beneficial for the industrial sector because of its applicability in remote load monitoring and fault detection services, with no need for intrusive metering and expensive smart sensors.

Hart introduced the primary techniques of NILM in the 1990s to disaggregate the energy consumption of residential units [7]. Later, researchers introduced different methods for energy disaggregation and improved the primary disaggregation plan [8]. Following different methods like the Hidden Markov model (HMM) [9], graph signal processing (GSP) [10], and deep learning (DL) [11], several advanced NILM algorithms have been proposed. Before 2012, the top technique to extract features for the classification of images included hand-crafted detectors such as scale invariant feature transfer (SIFT) and difference of Gaussian [12]. In 2012, the AlexNet proposed by Krizhevsky et al. won the ImageNet Large Scale Visual Recognition competition with an error score of 15%, chased by the second algorithm with an error score of 26%. Krizhevsky et al.'s winning algorithm [13] did not employ a hand-crafted feature detector. As an alternative method, Krizhevsky et al. utilized a deep neural network (DNN) trained to automatically extract a features hierarchy out of a raw image. The proposed 1D CNN-RNN method is a DNN-based automatic feature learning method located in the artificial neural networks (ANN) field, as shown in Figure 1.

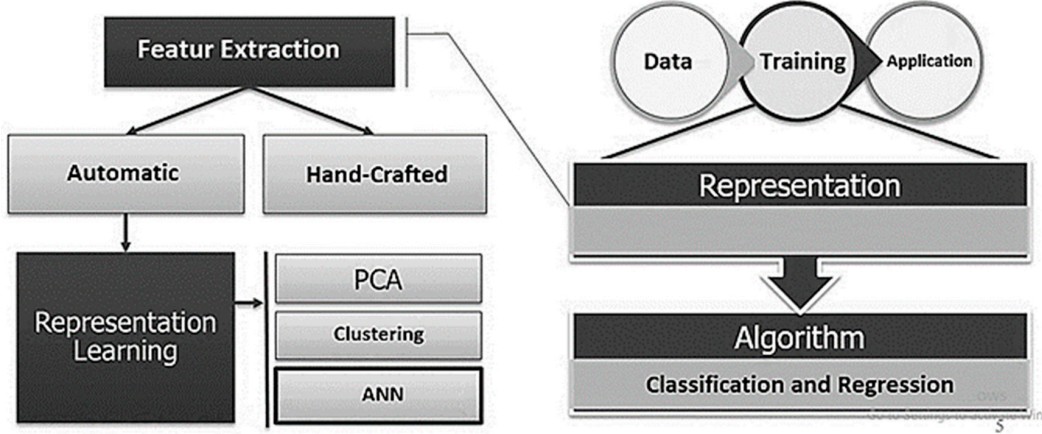

**Figure 1.** Taxonomy of proposed feature learning method located in ANN.

Currently, the use of deep learning is not limited to image classification and it is used in machine translation, automatic speech recognition, and even learning for playing computer games [14–16]. The present study examines the ability of using DNNs for energy disaggregation. Roos et al. (1999) [17] utilized NILM in small neural networks for the first time. The mentioned study was a proposal only, which was in [18] as well. However, a small neural network did not appear as a suitable choice to teach

recognition of the hierarchy features. A great innovation in image classification took place when the graphics processing unit (GPU) was introduced to compute power for DNN learning in a huge dataset. Apart from the traditional methods such as HMM, neural networks, and optimization, the DL methods are capable of deep feature learning for recognizing appliances with a high accuracy. The advantages of DL are automatic feature learning, multilayer feature learning, a high accuracy, a high power of generalization, and hardware and software support by Nvidia, Google. At this point, we examined DNNs in terms of having a desirable performance for energy disaggregation. The major contribution of this work is adding a hybrid functional API model to NILM in the area of DNNs.

Three algorithms were examined for each model and one DNN was trained for each target device. Two benchmark disaggregation algorithms (ConvNet and LSTM (long short-term memory)) were used to compare the disaggregation performance of the three models based on metrics. Moreover, generalizability of the proposed method was examined for appliances not seen in the training process because, eventually, when NILM is deployed at a scale running environment, we very infrequently have ground truth data of appliances for the houses for which the aim is to disaggregate. Therefore, it is important that NILM methods are able to generalize to unseen houses. It is notable that after training, the designed model of energy disaggregation does not require data of appliances of ground truth from each house. End users just need to acquire aggregate data. That is because each DNN model trains the extract of its target appliance so that it has the power of generalization to unseen samples of that appliance. In the same way, DNNs trained to perform the classification of images are trained on many category samples, such as cats, dogs, etc., and generalize to unseen samples of each category.

To provide more background, we will summarily sketch how presented DNNs could be deployed at scale. Each DNN undergoes supervised learning on many samples of its target appliance type so that each DNN trains to generalize fine to unseen appliances. The training process is very expensive and the processing, even with fast GPU, takes several days. However, processing the training more than once is not necessary, because after the training of deep neural nets, the model is ready to deploy on a server or edge device for real-time NILM and appliance classification. In addition, it can be trained on the 12 h free GPUs named Tesla K80 powered by Google Colaboratory. The inference is much cheaper when these DNNs are trained; it takes approximately a processing second per network of DNN for a week of aggregate data on a GPU. In this paper, the DL network is trained with Google Colab on the free Tesla K80 GPU [19] using Tensorflow backend. Colab is a research tool for machine learning (ML) research and education. A Jupyter notebook environment requires no setup to use [20]. From unseen houses, data would be fed to each DNN. Each network of DNN should extract the energy demand for its target appliance. This application would be computationally high-cost in terms of processing for a processor embedded in a smart meter. This processing would run on new STM32CubeMx.AI (3.4.0, STMicroelectronics, Geneva, Switzerland), which is a software tool for the fast mapping of pre-trained ANN models (generated using off-the-shelf DL frameworks) and is optimized for the STM32 Microcontroller family [21].

As an alternative, the aggregate data could be sent to the cloud from the smart meter. In the one 16-bit sampling of 0–64 KW in 1 watt steps, for every ten seconds, the data storage requirements are 17 KB per day in levels of uncompressed data. This data should be compressible because there many of periods in demand of domestic aggregate power with slight changes. The overall data storage requirements for one year of data from 10 M users would be 13 TB using a 5:1 compression ratio, and neglecting the date time index, which could not occur on two 8 TB hard drives. If a week of data can be processed in a second per home by using further optimizations, then 10 M user's data could be processed by 16 GPU; as another possibility, disaggregation could be operated on a processing device within each home, such as a laptop, mobile phone, or dedicated hub of disaggregation, which could process the disaggregation. This processing would run on a Raspberry Pi-based state-of-the art Energy Disaggregation using Intel's new Neural Compute Stick (NCS). The Intel Movidius NCS [22] promises to boost the rate at which the Pi can carry out heavy tasks like facial and object recognition. Although GPU makes processing faster, it is not required for disaggregation.

*NILM Background*

For a technical background of NILM, a basic NILM framework consists of three important stages: data acquisition (DAQ), feature extraction, and pattern recognition. Figure 2 describes these basic steps for an NILM platform.

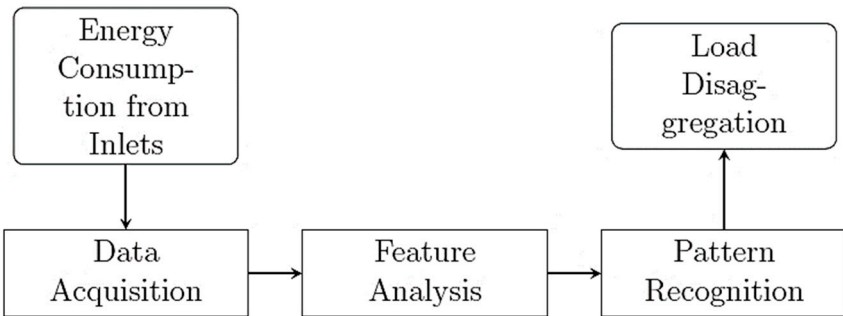

**Figure 2.** A schematic of NILM workflow.

Primarily, the energy information is gathered from the power inlets using a data logger or data acquisition device. Then, the collected data is analyzed based on the feature extraction algorithm. The objective of this phase is to diminish the dimensionality of the original dataset and smooth the waveforms of raw signals. The machine learning (ML) algorithm finally compares those extracted features with various appliance patterns in the database. As an example of research problems and gaps in demand side management (DSM) referring to [23–25] about DSM information, the main contributions of the study are the intrusive monitoring-based DSM framework comprised of advanced communication mediums such as a wireless home area network (HAN) and wireless internet access (ZigBee), and home energy management systems (HEMS) tested on a microgrid environment which consists of solar PV, wind, and back-up diesel generators. The proposed smart HEMS in DSM is a fuzzy logic-based load controller which optimizes household appliances based on the available renewable generation in the microgrid and local voltage measurements from the smart meter. The current challenge is focusing on NILM and supervised energy disaggregation based on deep feature learning (DFL). After training, the proposed deep neural model can be deployed on a smart meter like Figure 3, which is able to classify home appliances from the data of main meters with no need for expensive smart outlet sensors.

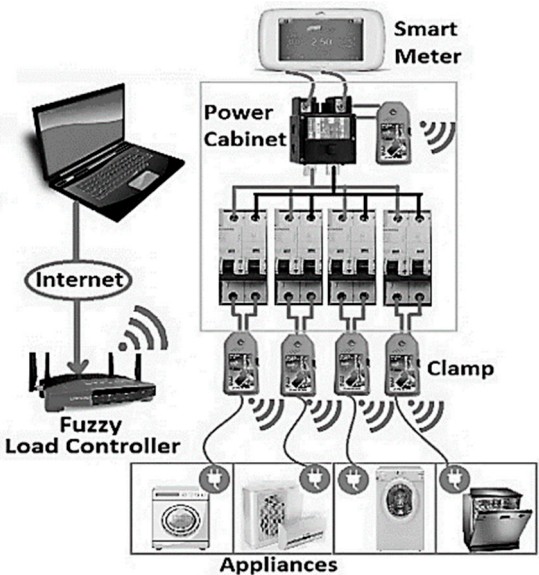

**Figure 3.** Architecture of the HEMS [23].

With the release of smart meters, the significance of effective NILM methods has risen rapidly. NILM predicts the energy consumption of individual appliances given their aggregate consumption; at this point, the contributions of the study are a review of the recent research that has provided novel insights into the deep neural networks (DNN) and the proposal of a more effective hybrid technique. In this way, an extreme learning machine approach to effective energy disaggregation is presented, which can only be monitored at a single, central point in the household, providing various advantages, such as a high accuracy and reduced cost for metering equipment. The structure of this paper can be summarized as follows: Section 2 presents some information about the Learning process and error function in artificial neural networks (ANNs); Section 3 provides an introduction to DNN and explains how to provide the training data for proposed energy disaggregation through DFL; Section 4 presents how three architectures of DNN are fitted to NILM; Section 5 expands the 1D CNN-RNN based on Metrics Tensorboard [26] and describes how to create disaggregation with a functional API model; Section 6 discusses the disaggregation results of the proposed model and the two benchmark NILM methods; and finally, Section 7 discusses the conclusions, and proposes future work about DL on the Edge for NILM.

## 2. Learning Process and Error Function

The learning process in ANNs, also called the training process, is mostly a function of pattern recognition. Like human beings, ANNs learn based on examples so that they learn about their environment by adjusting synaptic weights and the bias level. There are two types of ANN learning; supervised and unsupervised learning. The former happens under the supervision of a teacher, i.e., a programmer. When the teacher sets the outputs (also called target), learning is a function of the response of ANN to teaching and adjacency of the actual output to the expected output (target). The latter happens without the presence of a teacher [27,28]. In this work, supervised learning is adopted. Semi-supervised learning is a combination of tagged and untagged data that is used for the training process and in general, untagged data are far more common than tagged data. This is the most suitable method for NILM [29]. It is notable that practical aspects (not only the theoretical aspects) of NILM are confusing for individuals without knowledge in this area. When supervised and unsupervised learning is used for machine learning, the training/testing data should be of a similar range. For energy disaggregation in practice, supervised learning is used for training (aggregated and sub-meter data are used) and unsupervised learning is used for testing on houses that are not part of the learning process (only aggregated data is used).

In addition, semi-supervised learning assumes that tagged and untagged data are from the same range; while in NILM, the data represent different houses, i.e., different ranges. In addition, while both tagged and untagged data are used for training, the testing phase only uses untagged data from the houses not included in the learning phase. Thereby, the issue of terminology is not concluded and needs further consideration. Computation of feed forward through the training process is always done using error signal feed backward, which represents a learning network. The concept of training NN (neural networks) is the same as the definition of network error. Rumelhart and McClelland defined a term of error that related to the difference between the actual output and target outputs [30]. The actual output is yielded by feed forward computations. The term of error represents a success measure of training the network with a specific training set. For minimizing the patterns of total error in the training set, the gradient descent is applied, and weights are modified in proportion to the negative of an error derivative as in the following equation:

$$\Delta w_{ji} = -\eta \left[ \frac{\theta E}{\theta_{ji}} \right] \tag{1}$$

where, $\eta$ represents the learning rate (LR) and $E$ represents the total error. At the next step of the algorithm, the output signal of network, which is now part of the training data, is compared with

the target output. The difference between the actual and target outputs represents the error of the output layer neuron. The signal is fed backward for weights adjustment. The mean square error (MSE) function [31] is the most common error function used in the backpropagation learning algorithm. The error is computed using the gradient descent method and it is used to minimize MSE between the actual output and target output for all the probable inputs. The MSE function is defined as

$$E = 1/2 \sum_k \left( t_{kj} - O_{kj} \right)^2 \tag{2}$$

where, $t_{kj}$ is the value of the target from output node $k$ to the hidden node $j$ and $O_{kj}$ is the actual output from the output node $k$ to the hidden node $j$.

## 3. Deep Learning

Deep learning (DL) is a specific set of machine learning (ML) methods using ANN, which is aspired, to some extent, by neuron structures of the human brain. In a less formal sense, the term "deep" refers to several layers in the ANN; however, this meaning has changed over time. Four years ago, 10 layers were enough to assume a network deep, while today, deep networks are comprised of hundreds of layers. DL is an actual game changer in ML area as it has used a few smart methods successfully in many different fields (speech, image, text, audio, and video) and displaced the boundaries. The success of DL lies with access to more training data (e.g., ImageNet in the case of images) and less expensive GPUs for highly efficient digital computations. Facebook (Menlo Park, CA, USA), Google (Menlo Park, CA, USA), Microsoft (Redmond, WA, USA), Apple (Cupertino, CA, USA), Amazon (Seattle, WA, USA), and many other companies use these DLs to analyze a large volume of data. The technique is no longer limited to front end scientific researchers and it has become a common practice in large industrial firms and an integral part of modern software. This indicates the necessity of mastering skills in this area.

### 3.1. Convolutional Neural Networks

Convolutional neural networks (CNNs) have started a deep revolution in DL given their capacity for processing image-like data using local connections, shared weights, pooling, and several layers [32]. Although the main influence of CNNs has been on machine vision and they use 2D images as inputs, this paper uses a 1D CNN since the data under consideration are a set of 1D input vectors. Figure 4 is a graphical illustration of the CNN sequence. Each layer is explained later.

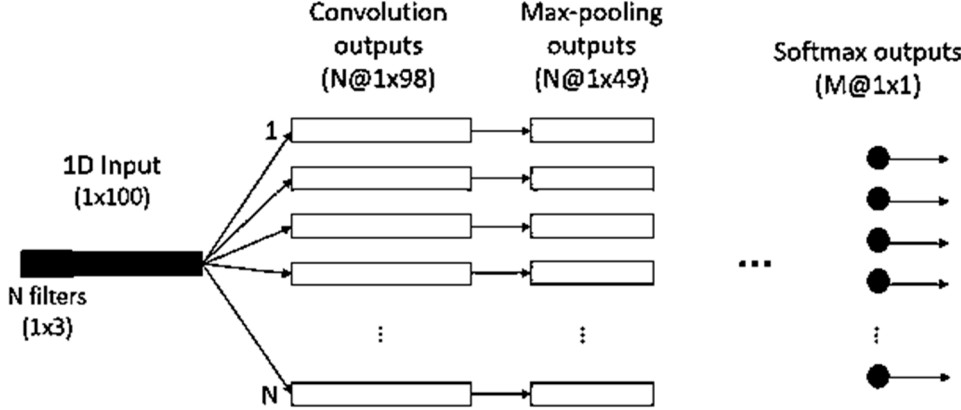

**Figure 4.** Representation of 1D CNN architecture.

The input is supplied through a series of convolution layers followed by a max-pooling layer that form conventional deep structures. "N" represents the number of filters on each layer. One "stride 1" and one "stride 2" are taken into account for convolution and max-pooling stages, respectively.

Without padding, the input sequence is shortened. The classification output is a softmax [33] layer with M outputs with the same number of batches. To gain a better understanding of the main layers constituting CNNs, two examples are pictured in Figure 5. Figure 5a illustrates the filtration of convolution layers and Figure 5b illustrates the max-pooling process (also called secondary sampling).

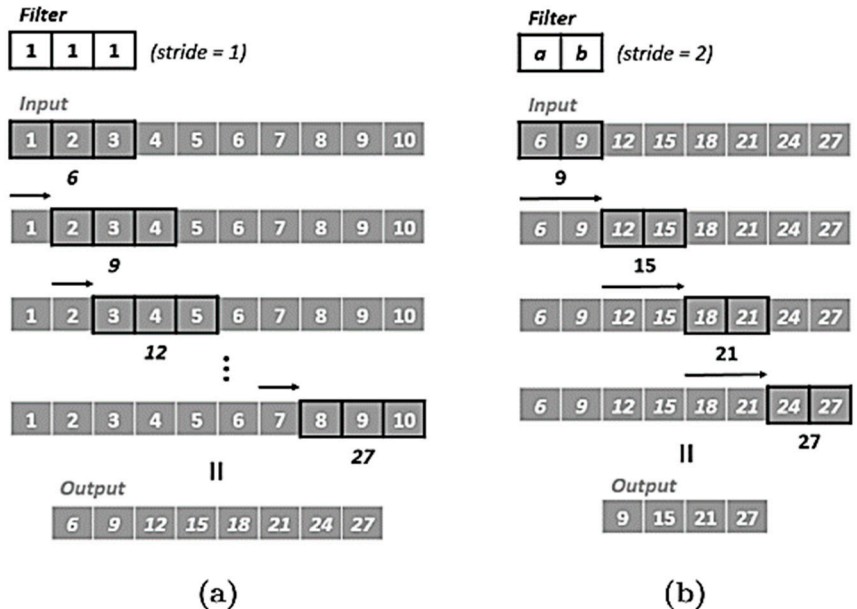

**Figure 5.** (**a**) Illustrates the filtration of convolution layers and (**b**) Illustrates the max-pooling process in 1D CNN.

The term "max-pooling" is a better choice as it includes the max function (a, b). Moreover, there are more sub-sampling functions like mean pooling. Each CNN layer contains a set of filters to extract connected local information and transfer it to the next layer. Therefore, weights from the previous layer are connected to weights to the next layer. In this way, CNNs can recognize features that remain unchanged throughout translation and more details are added to them with an increase in the depth of the network [34]. The reason for this is that CNNs have gradually replaced feature extractors engineered by humans, including those used to determine audio patterns.

After a stack of convolution and pooling layers, there can be one or several fully connected layers before implementation of the softmax function. This is equal to adding one MLP to the end of CNN. In the case of the softmax activation function, the final layer needs an outputs count equal to the number of classes. For multi-class problems, the last layers perform the softmax operation, so that the output is squashed to (0,1) and the sum of outputs will be equal to 1. In this way, the classification output can be assumed as a probability measurement with the following formula:

$$g(b)_{j=\frac{e^{b_j}}{\Sigma_{k=1}^{K} e^{b_k}}} \tag{3}$$

This equation consisted of the $j \in RK$ output from softmax activation function $g$; that is, with a $k$-dimensional vector called $b$, it is connected to the output layer. Although, softmax is an activation function like the rectified linear unit (ReLU) [35], the derivative has a unique feature of dependence to the output index, as the equation is computed for all $j$.

Figure 5a shows a filter with input vectors and one convoluted stride 1; that is, the filter is shifted one stage by one stage toward the input vector. Afterwards, the central value is updated to be equal to the weighted sum of the input section and weight of the filter. Figure 5b shows that the max-pooling filter, i.e., max (a, b), is executed on the output layer of convolution with stride 2 so that the output vector is obtained with one-half length.

### 3.2. Recurrence Neural Networks

When the backpropagation algorithm was introduced, the most exciting application envisioned for it was in RNNs training. These networks are widely different from multilayer perceptron (MLP) and CNNs as they have a dynamic memory that can be altered through recurrent connections. Studies have shown that, thanks to this feature, recurrent networks have a better performance compared to feed forward neural networks (FFNN) with regard to temporary dependent signals. Cleary, recurrent neural networks (RNNs) contain a very deep FFNN with layers of the same weights [36]. This is clearly demonstrated in Figure 6, where an RNN network is unfolded through time. It was through the unfolding that Backpropagation Through Time (BPTT) was developed for RNN. As the name implies, by taking RNN as an FFNN, it is possible to perform backpropagation. However, although FFNNs have different parameters that need to be updated for each layer, RNNs have identical parameters for each temporary layer. Although RNNs are the same as deep FFNNs in terms of performance, they suffer from the same problems that we see with large networks, like vanishing/exploding and loss of gradient.

Given the backward flow of gradient computations through time, using the conventional BPTT causes a tendency to develop high or low values so that the network incurs the cost of vanishing or exploding. Error through time depends, in an exponential way, on the weights. Therefore, when the gradient is explosive, oscillating weights are inevitable. On the other hand, when the gradients vanished, slow learning or failure is inevitable [37].

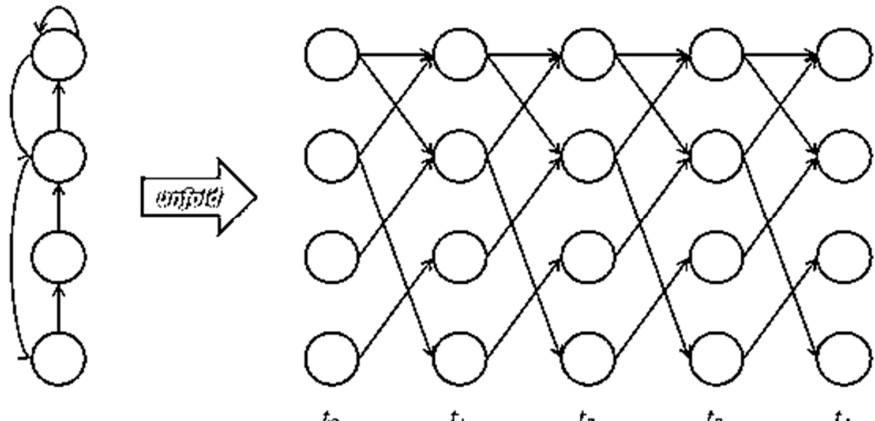

**Figure 6.** Illustration of RNN unfolded through time.

On the left, the network architecture is pictured and the arrow represents the recursive connection. On the right, the connections are pictured in a space where each time step represents a new layer.

## 4. D CNN and RNN on NILM

This section discusses the theoretical foundations of the experiments to solve at least some of the problems discussed above. In their study, Kelly and Knottenbelt [38] used DL techniques to solve the NILM problem.

### 4.1. REDD Dataset and Preprocessing

The reference energy disaggregation dataset (REDD) is a free database on NILM that contains accurate information about the power use of several houses. Its purpose is to expand research work on energy disaggregation (to determine the share of home appliances of aggregated electrical signals).

The data are specially designed for energy disaggregation work, which means determining element appliances connected to an aggregation electrical signal. The REDD is comprised of the total power use of a house and the power used by a special circuit/device for several houses over several months. For each monitored house, the following items are recorded: *a-* total electrical signal of the

house (current monitors at the both power phase and voltage monitor at one phase) recorded at a high frequency (15 KHz); *b-* up to 24 unique circuits at a house, with each tagged with home appliance categorization at 0.5 Hz; *c-* up to 20 outlet level monitors at a 1 Hz frequency. The main emphasis is on input electronic devices where several devices are connected on one circuit. Examples of such data are pictured in Figure 7. On the 15 June 2011, 10 houses were monitored and the data represents 119 days (combined for all houses), 268 unique monitors, and more than 1TB unprocessed data.

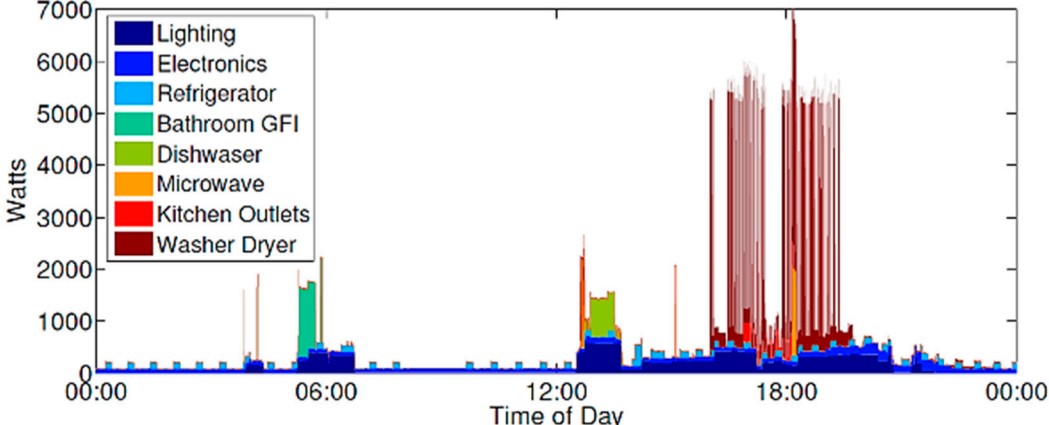

**Figure 7.** Energy consumption over a day for one of the houses in REDD [38].

As noted in [39], REDD is not free of drawbacks. Therefore, before feeding the system with data, it should be preprocessed. The absent points were "absent range" is less than 20 s, are filled using the forward filling method (otherwise a void in the data is unavoidable), and the data is sampled through computing the mean points with a 4-s time gap. To create mini-batches for the neural network, the ranges with a gap (ranges higher or lower than 4-s) are omitted. Through this, filters at convolution layers always look for samples with the same gap, which is very important; otherwise, filters will learn different things with different attempts. All the inputs are modified to yield a mean value of zero. The data scale is gradually decreased by a fix factor of 500, which helps the primary value assignment. It is notable that this is not essential at the batch normalization stage, since the batch normalization ratio to input value remains constant with a change in the scale (BN (Wu) = BN (($\alpha$W) u).

### 4.2. Combining CNN and RNN

Since 1D methods of convolutional networks process the input batches directly and without dependence, they are not affected by the sequent of time stage (beyond local scale, convolution windows size), despite RNNs. However, it is possible to aggregate several convolutional and aggregation layers to recognize long-term patterns, so that the upper layers will consequently experience longer pieces of the main inputs. It is notable, however, that these methods are not reliable enough to induce sequence sensitivity. A strategy to combine the speed and agility of convolutional layers with the sensitivity of RNNs to the sequence is to use a 1D CNN (see Figure 8) as a preprocessing stage before the implementation of RNN (see Figure 9). This strategy is useful for series that are too long to be practically processed by RNNs (e.g., sequences with thousands of steps). The convolutional forms of long input sequences, through this, are converted into very short sequences (down sampled) with a higher level of features. The sequence of the extracted feature is converted into the input for the RNN section.

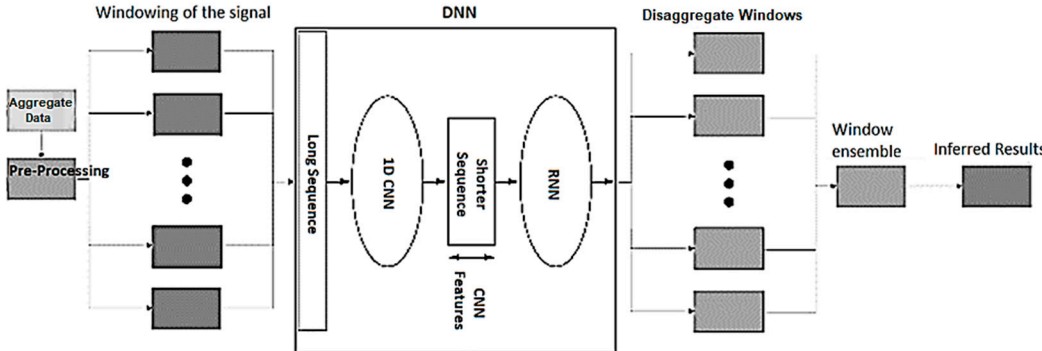

**Figure 8.** Flowchart of the system operation.

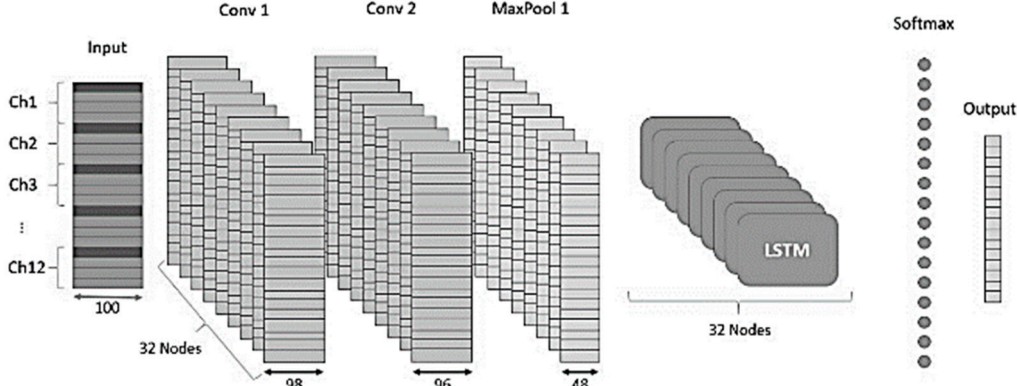

**Figure 9.** Schematic of CNN-RNN model.

### 4.3. Long Short-Term Memory (LSTM)

Long short-term memory (LSTM) is an RNN, which was first introduced in 1997 by Hochreiter and Schmidhurber [40]. The architecture is commonly used to solve vanishing gradient problems in vanilla RNNS [41]. The LSTM uses gates for better control of gradient flow. However, the error is trapped in the memory at the backpropagation state and creates the known effect of error carousel [42]. This problem was solved to some extent after the introduction of the peephole connection by Gers et al. [43], so that accuracy of the model was improved. Gers et al. also introduced forget gates; the gates resulted in a considerable improvement in performing several tasks featured with a computational operation. Using these gates, LSTM can learn the irrelevant contents of the memory until local automatic reset.

The input in Figure 9 is first fed to two convolution layers followed by a maxpool layer. The extracted feature vectors by these layers constitute the input of RNN with 32node LSTM. The output size of each layer is determined under the layer. At the end, a softmax layer with 16 nodes is added, which is not pictured in the schematic view. However, the outputs of the last maxpool layer are accumulated on a big vector before being fed to the output layer.

### 4.4. Metrics for Evaluating the NILM

There are different metrics that can be used to assess NILM methods, which also makes it hard to compare assessments based on different methods and algorithms for load monitoring. At first, when the algorithms are implemented with two modes (ON/OFF), the metrics of assessment are based on the percentage of correct load classification to notable change in total used power. There are several metrics available for this purpose and before introducing them, some of the variables need to be defined:

*TP* (total number of real positives): When both the device and ground truth are ON.
*FP* (total number of fake positives): when the device is ON and ground truth is OFF.

*TN* (total number of real negatives): when both the device and ground truth are OFF.
*FN* (total number of fake negatives): when the device is OFF and ground truth is ON.
*P*- Total number of positives on ground truth.
*N*- Total number of negatives on ground truth.

These variables are positive when power usage is above the threshold and they are negative when power usage is less than or equal to the threshold, i.e., $y(t) \leq \gamma$ where $y(t)$ stands for power usage in a time frame and $\gamma$ is the threshold. The threshold for each device is set manually. This indicates that when the device is ON, it is not considered as "standby", although it cannot measure all the aspects and this makes the algorithm imperfect. Therefore, it is always better to use more than one metric to measure the capacity of the algorithm. Some of the most common metrics used in this regard are discussed below.

### 4.4.1. Proportion of Total Energy Classified Correctly

The proportion of total energy classified correctly (P.T.E.C.C) metric is widely used as it is compatible with many monitoring methods. This measure pays special attention to devices with high energy use, which is a good feature as it is important for correct classification and the control of total use. In scenarios where users do not have control over some devices, such devices constitute a small part of energy use; therefore, this metric remains a good measure, even with incorrect classification. When all the available home appliances are taken into account, P.T.E.C.C. can be written as follows:

$$Acc = 1 - \frac{\sum_{t=1}^{T} \sum_{i=1}^{n} \left| y_t^{\wedge (i)} - y_t^{(i)} \right|}{2 \sum_{t=1}^{T} \overline{y}_t} \tag{4}$$

When only one device is taken into account, we have:

$$Acc = 1 - \frac{\sum_{t=1}^{T} |y_t^{\wedge} - y_t|}{2 \sum_{t=1}^{T} y_t} \tag{5}$$

where, $y_t^{[t]}$ and $y_t^{\wedge (i)}$ represent the actual and estimated energy use by the *ith* device at *tth* time, respectively and $y_t^{\wedge} = \sum_{i=1}^{n} y_t^i$ is the total energy use at *tth* time. It is notable that this metric is different from the method proposed by Kelly and Knottenbelt (referenced at Section 4.1). They wrote the total energy use of all home devices as a denominator of the fraction and total energy use of one device as a numerator. However, since we assess each device separately, the actual energy use of the device is writing at the denominator.

### 4.4.2. Mean Normalized Error

Mean normalized error is the relative error in the energy dedicated to each device through time.

$$Acc = \frac{\left| \sum_{t=1}^{T} y_t^{\wedge (i)} - \sum_{t=1}^{T} y_t^{(i)} \right|}{\sum_{t=1}^{T} y_t^{(i)}} \tag{6}$$

### 4.4.3. Recall

With regard to the disaggregation of energy, this metric is a part of energy that is classified and measured correctly.

$$recall = \frac{true\ positives}{true\ positives + false\ negatives} \tag{7}$$

### 4.4.4. Precision

With regard to energy disaggregation, this metric represents what percentage of the total energy assigned to a device is actually used by that device.

$$precision = \frac{TP}{TP + FP} \tag{8}$$

### 4.4.5. Accuracy

The ratio of real results in all cases.

$$accuracy = \frac{TP + TN}{P + N} \tag{9}$$

### 4.4.6. F1 Score

Harmonized mean accuracy and recall.

$$recall = 2 * \frac{precision * recall}{precision + recall} \tag{10}$$

### 4.4.7. Mean Square Error

Mean square error is one of the most common metrics used in ML and minimizing it leads to several statistical advantages.

$$MSE = \frac{1}{n} * \sum_{j=1}^{n} \left(Y_i^{\wedge} - Y_i\right)^2 \tag{11}$$

Because *MSE* is a large number in most of the cases, sometimes, it is preferable to use the root mean squared error (*RMSE*) defined as $RMSE = \sqrt{MSE}$.

### 4.4.8. Categorical Cross-Entropy

$$L_i = -\sum_{j} t_{i,j} \, \log\left(p_{i,\,j}\right) \tag{12}$$

where, *i* stands for instantaneous time, *j* stands for status of the device, $t_{i,j}$ stands for target probability of a device at *j* at the time *i*, and $p_{i,j}$ is probability of a device at *j* at the time *i*. Categorical Cross-Entropy was used as a metric for training the network.

## 5. Houses not Seen During the Training for Testing

The proposed algorithm was tested on the houses that were not part of the training process (see Table 1). The model was trained using 80% of the data and disaggregation was carried out using the rest of the data. This method is highly similar to the validation against error test, which is normally used in ML. The following table lists the houses used for training and testing stages. It is notable that the tests are aimed at demonstrating generalizability of the system, which is the main goal of an ML model.

**Table 1.** Selected houses for training/testing.

| Device | Training | Testing |
|---|---|---|
| Microwave | 1, 3 | 2 |
| Dish washer | 1, 3 | 2, 4 |
| Refrigerator | 1, 3 | 2, 6 |

The results of this scenario are pictured in Figures 10–14.

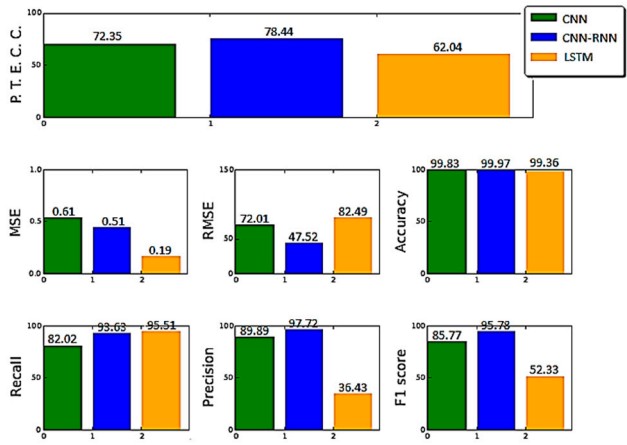

**Figure 10.** House 2 test on Microwave.

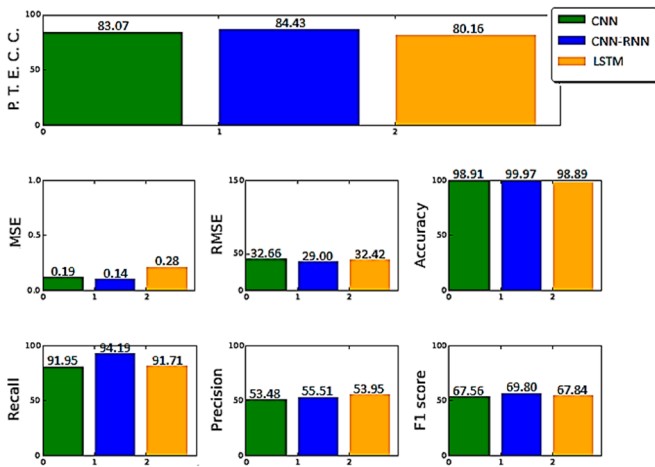

**Figure 11.** House 2 test on Dishwasher.

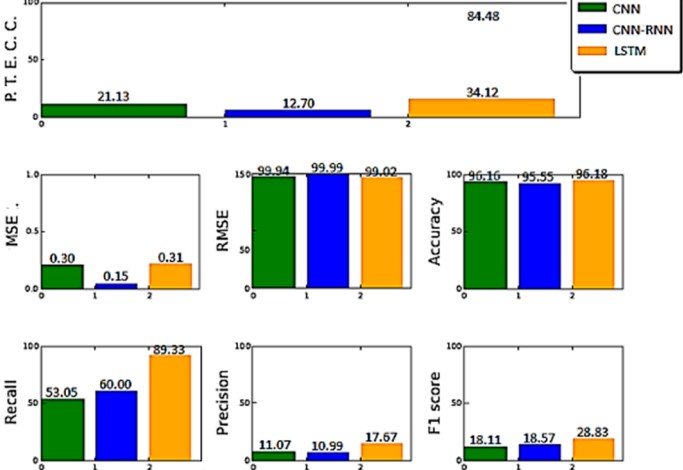

**Figure 12.** House 4 test on Dishwasher.

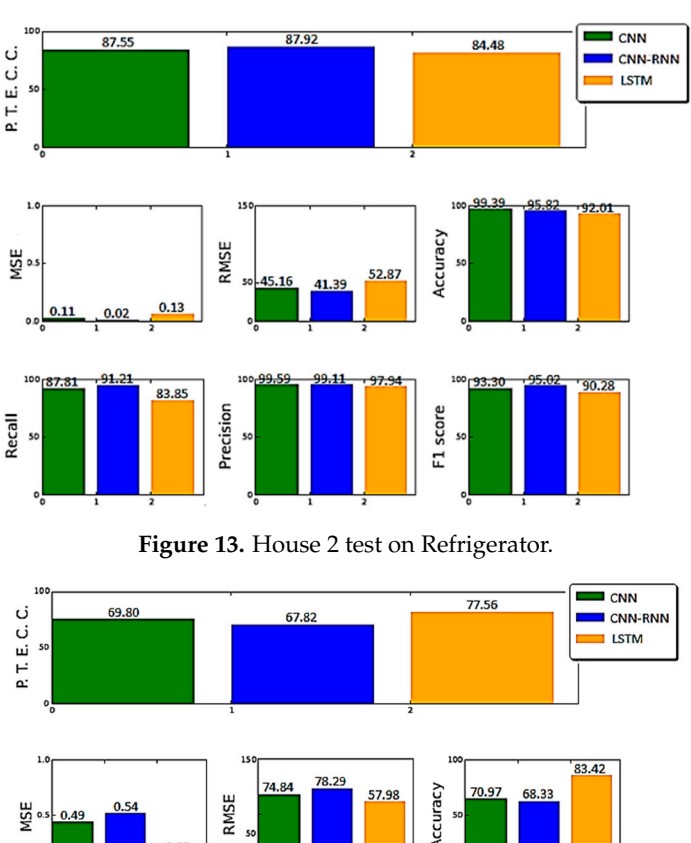

**Figure 13.** House 2 test on Refrigerator.

**Figure 14.** House 6 test on Refrigerator.

## 6. Results

The proposed model was suggested as an effective solution to guarantee energy disaggregation in smart grids. It was proposed to apply DFL to NILM. The main advantage of RNN DFLs over LSTMs and denoise autoencoders is that there is no need to fine-tune the whole network by the iterative back-propagation algorithm. This can quicken the learning speed and strengthen the generalization performance. The results of the previous sections showed the power of DFL on NILM. All the models introduced in the paper achieved good results, even when the disaggregation of houses was not taken into account. Based on the metrics of P.T.E.C.C, Accuracy, Recall, Precision, F1 Score, and MSE are widely used as they are compatible with many monitoring methods. The 1D CNN-RNN architecture had the best total performance on House 2 test on Refrigerator and the CNN network had the lowest performance. This indicates that specially designed architecture is very important for performance, even in NN of the same type.

In the case of microwave sets, none of the networks demonstrated a considerable performance and in the case of dishwashers and refrigerators, this indicates the complicated temporary behavior of these machines. Although RNNs have a good performance with regard to temporary patterns, the length of the big window used for disaggregation of the refrigerator may jeopardize optimization. Therefore, any improvement in this regard my lead to a higher performance. The results for house No.2 were better than other houses. This might be due to the fewer number of circuits in this house (11 circuits) compared with other houses (20 circuits; see REDD mentioned at Section 4.1). The results

showed how DNNs can be a good choice for NILM. The final simulation using Google Colab Tesla K80 GPUs took one day.

The advantages of this study are as follows:

- Implementation of DL in NILM and its potential for problem solving were examined.
- A combined method was introduced to show the approach of implementing DL methods using a small amount of real data.

## 7. Conclusions

Energy and sustainability issues raise a large number of problems that can be tackled using approaches based on data mining and machine learning; however, traction of such problems has been slow due to the lack of publicly available data. To test the capability of DFL on NILM, we conducted a series of experiments on the standardized UK Domestic Appliance-Level Electricity (UK-DALE) dataset. Notably, the amount of training data in this dataset is limited in comparison to that used for speech and image recognition. This permits us to prove that DFLs are indeed a viable solution to energy disaggregation when the amount of training data is limited, which hinders the training and generalization capabilities of state-of-the-art deep models, such as CNN and LSTM. The REDD is the data that can be downloaded from an initial version of the data set, containing several weeks of power data for six different homes and high-frequency current/voltage data for the main power supply of two of these homes. The data itself and the hardware used to collect it were described thoroughly. Those wishing to use the dataset in academic work should cite REDD paper as the reference. Although the data set is freely available, for the time being, they still ask those interested in the data to email (kolter@csail.mit.edu and mattjj@csail.mit.edu) and receive a username/password to download the data. This paper used the REDD, and we discussed the DL approach to disaggregation and presented benchmark disaggregation results using the DL technique.

## 8. Future Works

The solution proposed in this study still needs improvements. The development of combined models is a good option for several problems like speech recognition. Therefore, combined models are good options for NILM. It is important to study better Adversarial samples in NILM and create more stable NNs to work with such samples. It is notable that the models used here can be improved through using a group of models. Moreover, the objective of the study was to show the potential of DNN in NILM, not to obtain the best results. NILM can be cost effective, especially when DL is working on a low-cost embedded platform. Ever since the uTensor [44] project started, a microcontrollers (MCUs) artificial intelligent (AI) framework, AI on MCUs has enabled cheaper, lower power, and smaller edge devices. It reduces latency, conserves bandwidth, improves privacy, and enables smarter applications. uTensor is an extremely light-weight ML inference framework built on Mbed and Tensorflow (see Figure 15). It consists of a runtime library and an offline tool. The total size of graph definition and algorithm implementation of a 3-layer MLP produced by uTensor is less than 32 kB in the resulting binary (excluding the weights).

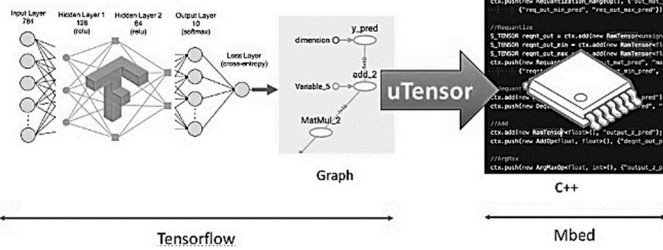

**Figure 15.** AI inference library based on mbed and TensorFlow.

As future work, we want to introduce the DL on the Edge devices for NILM on a low-cost embedded board using a novel uTensor inference library that should support any Mbed enabled board. uTensor is young and undergoing rapid development.

**Author Contributions:** İ.H.Ç. and V.F. contributed to the design and implementation of the DNN based energy disaggregation methodology, to the achievement of the results and to the writing of the manuscript.

**Funding:** This research received no external funding.

**Conflicts of Interest:** The authors declare no conflict of interest.

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
