# Peer review of "New Design of a Supervised Energy Disaggregation Model Based on the Deep Neural Network for a Smart Grid"

_energies, doi:10.3390/en12071217_

Round 1

Reviewer 1 Report

The authors proposed a supervised energy disaggregation model based on deep neural network for smart grid. However, the contribution of the paper was not addressed clearly an there are many flaws the authors need to rectify to make the paper publishable.

- The paper is not well-written and well-presented.

- Research problems and gaps were not identified correctly. Need to critically review the existing literatures to identify the research gaps and accordingly objectives of the research. Authors may go through the following papers:

    Implementation of advanced demand side management for microgrid incorporating demand response and home energy management system

- Introduction section need to be modified addressing background, significance and contributions of the research.

- Section  2 contains very general well-known information about NN which may be reduced significantly. Those information will not carry any valuable message to the research.

- Section 3 is also very details. Authors may consider to merge Section 2 and 3 together including all required/summarised information related to the paper. May be you can start with DP.

- Few of the figures are really difficult to read.

- Results section should be elaborated and improved significantly with the findings and causes of the findings.

- What is M. N. E. in Figure 12 or so on, is it MSE?

- Results and conclusion section is not convincing. Need to highlight the contribution of the paper.

Author Response

The answer file available in attach.

Reviewer 2 Report

The paper propose an hybrid model of Energy Disaggregation through Deep Feature Learning for Non-Intrusive  Load Monitoring (NILM). Despite the topic is very interesting i have some concerns about the work.

-It is very difficult to identify the final objective of the study, the introduction does not define it. In addition, it is also difficult to have a clear picture or learn from the state of the art for the problem studied. I would recommend the authors to include a paragraph describing the purpose of the study and a section of related works.

-Although the methods are well described, they are not presented correctly in the field of energy with adequate references.

- The experiments carried out and the data used in them are not clearly described (for example, are they available? Why are they sufficient to support your results?) in the paper.I would recommend to the authors to reorganize the work to improve this aspect.

-The results presented by the authors claims to prove that their method is relevant, however I think that only one dataset it is not enough to prove that. I recommend the authors to extends their experiments with other datasets.

- A conclusion section is need it. 

Minor:

line 13: Load Monitoring (NILM) is able -> Load Monitoring (NILM), which is able

Author Response

The answer file available in attach.

Reviewer 3 Report

*Goal: Model of Energy Disaggregation through Deep Feature Learning for Non-Intrusive Load Monitoring to classify home appliances from the information of Main Meters.

*Problem (technological) to solve: smart outlet sensors are expensive.

*Discussion:Energy dissagregation methods.

Questions:

1) Explain better why you use DNN over other paradigms. Is it related to the concept of target Appliance ? why is it not necessary processing the training more than once ?

2) Introduction to ANN should be reduced and focussed on adaptative algorithm. Take care of the missing caption of figuer 4, in page 199. This figure is didactic, but its usage in the paper is limited. I suggest to erase it. 2.4-5-6 can be gathered in one section.

3) Deep learning is perhaps the most important issue to be summarized and explained in the present work. Pay attention to the written text, for instance in line 338 a reference is missing [].

4) Figure 9 should be enlarged. Again in lineas 359 to 360, we fing a proof of lacking formalism, units of Hz have to be formated adequately Also in line 367: 4-second time gap. 

5) Fig 10 is a little bit fuzzy. enhanced or make it clearer: resolution. Revise the whole paper in the same contaxt of units and clarity of images.

6) when you discuss the advantages in line 504 you cite small amount of data. At whta extent are results significative.

Thanks a lot for addressing questions.

Author Response

The answer file available in attach.

Round 2

Reviewer 1 Report

The authors put efforts to improve the paper though there are still few areas that need to improved. Contributions/novelty of the paper should be clearly define.

In page 4, authors highlighted paper including the title of the paper, however authors have not cited the paper in the reference. Do need to write the title of the paper, only reference is fine. Authors also need to reviewed other relevant research to highlight the novelty of the approach made by the authors.

Improve the quality of the figures.

Better to proof-read the paper with a professional proof-reader.

Need to put more attention in finalising the Abstract, Introduction and Conclusions section.

Author Response

The file is available in attach.

Reviewer 2 Report

I would like to thank the authors their efforts in order to answer my comments point by point. However, I think that the paper should be improved in some aspects before to be suitable for publication in the journal. 

- I still feel that the study has several limitations in the experiments. There is only one dataset, and what is more, this dataset present some issues (all the houses used had only one device of the same type). Due to this, it is very difficult to evaluate the relevance of the research.

- The paper still have to improve in its organisation. The should be a results section, with includes the results and the discussion and Conclusion and future works section, which summarizes the main achieves of the research and the future works.

Minor:

- In the section 4.4.3 the Recall formula is missing. In the rest of the presented measures is described. 

Author Response

The file is available in attach

Reviewer 3 Report

Dear authors,

I would like to read v2 of the paper with more readable figures and with a shorter section (eventually erasing this section) regarding neural networks. Indeed, explanations for this paradigm does to provide useful knowledge.

Regards

Authors should make figures clearer. Poor quality for images.

Author Response

The file is available in attach

Round 3

Reviewer 2 Report

I would like to thank the authors that have addressed the issue that I have suggested.

I have no more comments.

Author Response

The minor revision is available in manuscript. 

Reviewer 3 Report

Tahnk you for addressing the questions.

Author Response

(The authors gave the same response as above.)
